

# Towards impact-based flood forecasting and warning in Bangladesh: a case study at the local level in Sirajganj district

Fabio Sai[1,3,4], Lydia Cumiskey[2,3], Albrecht Weerts[3,5], Biswa Bhattacharya[4], Raihanul Haque Khan[6]

[1]A.I.Po, Interregional Agency for the Po River, Milan, 20124, Italy
[2]Flood Hazard Research Centre, Middlesex University, London, NW4 4BT, UK
[3]Deltares, Boussinesqweg 1, Delft, 2629 HV, The Netherlands
[4]Hydroinformatics Chair Group, IHE-Delft Institute for Water Education, Delft, 2611 AX, The Netherlands
[5]Hydrology and Quantitative Water Management Group, Department of Environmental Sciences, Wageningen University
   and Research, Wageningen, The Netherlands
[6]RIMES, Pathumthani, 12120, Thailand

*Correspondence to*: Fabio Sai (fabio.sai@agenziapo.it)

**Abstract.** Impact-based forecasting and warning services aim to bridge the gap between producers and users of warning information by connecting and increasing synergies between the components of effective early warning systems. We tested
qualitatively whether a warning message based on colour codes is understandable and useful to trigger risk mitigation actions at the local level in the flood-exposed communities of Rajapur and Ghorjan unions in Sirajganj district, Bangladesh. With a community-based approach for different groups of users (i.e. sectors), flood-impact scenarios were determined from past events and related to colour codes. These were developed into impact-based forecasting and warnings that can connect water levels, through the colour code, to localised guidance information tailored to sectors' needs on how to respond to the
expected flood. This approach was tested through a limited number of focus group discussions and interviews at the community level. Overall, the colour coded impact-based warnings were found to be an easy and understandable way to link water level forecasts to the necessary risk mitigation actions, however, further investigation is needed to validate these findings under real-time conditions. IBFW has huge potential in Bangladesh but its integration requires significant institutional changes, such as an inter-facing agency (long term) or team (short term), adjusted policy frameworks (standing
orders on disasters), and new resource allocations for skills development and technological innovation from national to local levels. Overall, this paper aims to offer a first insight into impact-based forecasting and warning services in Bangladesh to trigger further research and project developments.

## 1    Introduction

Bangladesh is one of the most vulnerable countries to floods, often resulting in high socio-economic impacts (Haque and
Zaman, 1993). The reason lies in the combination of the poor socio-economic features with the unique geophysical location in the vast floodplain at the confluence of three major rivers forming the Ganges – Brahmaputra – Meghna basins (GMB)



(Islam *et al.*, 2010). Particularly, Sirajganj district, located on the bank of Brahmaputra, has experienced high damages caused by severe flood events in 1988, 1998, 2004, 2007 and 2010 (Bhuiyan and Al Baky, 2014). The Jamuna River, the main branch of the Brahmaputra River bounding the eastern side of the district, is a braided river forming the so-called *chars* (islands in Bengali), which are unstable and flood prone morphological formations. Nevertheless, *chars* are home to some of

the poorest and most vulnerable people in Bangladesh (Paul and Islam, 2015). Traditional embankments are ineffective risk mitigation measure (Hossain Md. Z. *et al.*, 2008) because of the dynamic shape of *chars*. Therefore, policy interventions targeted to cope with the yearly floods are necessary to reduce adverse impacts (Haque and Jahan, 2015).

Early Warning Systems (EWS) are risk mitigation measures targeted to deliver information on an emerging hazardous event, to enable actions in advance that reduce the risks involved (Basher, 2006). A complete and effective EWS comprises of four

inter-related elements: a) risk knowledge, b) monitoring and warning service, c) dissemination and communication and d) response capability. A failure in one of these components determines the failure of the whole system (UNISDR, 2006). Risk knowledge refers to the combination of hazards and vulnerability at specific locations through systematic data collection and analysis targeted to understand the dynamic nature of risk. The monitoring and warning service includes the scientific aspects required to timely forecast the hazard and prepare accurate warnings in space and time. Dissemination and

communication ensure warnings reach all those at risk, and provide clear, simple and useful information through different channels. Response capability makes sure that warnings are received and understood activating all the actions needed to reduce risk (UNISDR, 2006). The World Meteorological Organization (WMO), conveying the outcomes of the Third UN World Conference held in Sendai (UNISDR, 2015), has highlighted the need for impact-based forecasting and warnings (IBFW) services to bridge the gaps hindering effective EWSs (WMO, 2015). The term 'impact-based' aims to translate the

hydro-meteorological forecast by shifting the paradigm towards end users, which is forecasting the expected consequences of hazards for different sectors of interest. Although recently introduced, best practices can be found in national meteorological services, like the UK Meteorological Office (UK MetOffice, 2017) and the United States National Weather Service (US NWS, 2017), as well as in international programs leaded by WMO through dedicated workshops (WMO, 2017).

In order to steer EWS towards impact-based services, Table 1 gathers the indicators mostly suitable for this purpose. The

indicators were selected from the 'Early warning checklist' (UNISDR, 2006) and supported by further literature.

**Table 1. Indicators identified for each of the four components of effective EWS (UNISDR, 2006), important for implementing IBFW services.**

| EWS component | Indicator | Description | References |
|---|---|---|---|
| Risk knowledge | Local risk assessment | Interaction of vulnerability and hazard scenarios for determining risk of the exposed elements with a detailed resolution. | Scolobig *et al.*, 2012. |
| | Hazard mapping | Hazard maps for different scenarios need to be developed to identify exposure for different hazard magnitudes. | Kwak *et al.*, 2015; Fakhruddin *et al.*, 2015; Koks *et al.* 2015. |





| | Vulnerability mapping | Vulnerable elements and critical infrastructures need to be mapped, documented un updated periodically. | Bhuiyan and Al Baky, 2014. |
|---|---|---|---|
| **Monitoring and warning** | Timely and accurate forecast | Good quality data have to be collected and processed in a real or quasi-real time to produce meaningful, timely and accurate forecasts. | Basher, 2006. |
| | Impact-based thresholds | Warnings must be prepared and issued based on expected impacts severity, enabling end user to take appropriate risk mitigation actions. | WMO, 2015. |
| | Geographic specific warnings | A dense monitoring network ensures a good coverage of the forecast being more specific, issuing warnings to localized targets. | Cumiskey *et al.*, 2015; Oktari *et al.*, 2014; Shah *et al.*, 2012. |
| | Sector specific warning | Warning content have be prepared and issued to different end users clusters with same needs. | Basher, 2006. |
| **Dissemination and communication** | Robust standing operating procedure | Government policy establishes the warning dissemination pathway and the roles for defining specific impact-based warnings. | Oktari *et al.*, 2014; Rahman *et al.*, 2012; Scolobig *et al.*, 2012. |
| | Complete and timely dissemination | Warnings must reach the entire exposed communities, including those in remote areas, in time before the hazardous event occurs. | Oktari *et al.*, 2014; Rahman *et al.*, 2012. |
| | Multiple dissemination channels | Exposed communities must be warned via different media according to their ability/possibility of using them. | Oktari *et al.*, 2014. |
| | End user's dissemination and communication needs | Message content, communication and dissemination means are tailored on end user needs, ensuring higher warning understanding. | Cumiskey *et al.*, 2015; Koks *et al.*, 2015; Fakhruddin *et al.*, 2015; Rahman *et al.*, 2012. |
| **Response capability** | Information on impacts and advisories | Messages must contain information on the expected impacts and the advisories on how to implement risk mitigation actions. | Shah, 2012; Basher, 2006. |
| | Community and volunteers education | Volunteers must ensure the coordination at the local level, helping communities to effectively respond to alerts. | Cumiskey *et al.*, 2015; Scolobig *et al.*, 2012. |
| | Preparedness and contingency plans | Hazard and vulnerability maps are used as a management tool for improving response and coordinating emergencies. | Cools *et al.*, 2016; van den Homberg *et al.*, 2016; Dewan *et al.*, 2014; Scolobig *et al.*, 2012. |
| **Cross-cutting component** | Local community participation | End users can actively contribute to all the four components of EWS. | Cools *et al.*, 2016; Fakhruddin *et al.*, 2015; Maidl and Buchecker, 2015; Dewan, 2014; Rahman *et al.*, 2012; Basher, 2006. |

In the Bangladesh context, the Department of Disaster Management (DDM) is responsible to collect and maintain risk information and assessments. Recently, DDM improved their risk assessment through the Multi-Hazard Risk and



Vulnerability Assessment Mapping and Modelling (MRVA) project. For flood risk, hazard was computed at national scale for different return periods (25, 50, 100 and 150 years) while vulnerability was defined as the combination of population, housing, livelihoods, critical facilities and infrastructures (DDM, 2016). The FFWC, under the Bangladesh Water Development Board (BWDB), is the agency entitled to monitor and forecast water levels (WLs). Currently the deterministic

forecast is available up to 5 days in advance at 54 representative stations on 21 rivers. Precipitation data, necessary to run the forecast, is collected from the Bangladesh Meteorological Department (BMD) and from Indian Meteorological Department (IMD) for the portion of the GMB basin lying in India (BWDB, 2014).

The FFWC provides warnings with respect to 'danger levels' (DL), fixed by BWDB for each station. A DL at a river location is defined as the level above which it is likely that flood may cause damages to nearby crops and homesteads

(FFWC, 2016). DLs are representative for large areas surrounding the forecasting stations, therefore floods might occur either for lower or higher levels at different locations. FFWC associates a green band for WL up to 50 cm below the DL, then yellow up until the DL, magenta for up to 1 m beyond DL and red for WLs more than 1 m above DL. FFWC issues flood forecast bulletins to government and non-government organisations (NGOs), media groups and other concerned parties. Bulletins contain observed rainfall, observed WLs and forecasted WLs for all the stations, emphasizing the expected

trends. Warning dissemination from the national to the local level is established and coordinated by the DDM through the Disaster Management Information Centre (DMIC). The main components of this framework are the Disaster Management Committees (DMCs) at different levels: national (National DMC – NDMC), district (District DMC – DDMC), sub-district (Upazila DMC - UzDMC) and local (Union DMC - UDMC). These committees are composed of both government and stakeholder representatives. Their responsibilities include communicating warnings from the FFWC through the

dissemination network and supporting the response actions (MoFDM, 2010). This network typically receives warnings from the DMIC communication system, via emails, phone calls and SMS. Furthermore, the information can be accessed by anyone via Interactive Voice Response service (IVR – service on request by calling 10941), FFWC website and in some cases on TV and radio. Union Digital Centres (UDCs) provide accessibility to such facilities also in rural areas (A2I, 2017). For a schematic diagram of the institutional network for warning dissemination see Shah *et al.* (2012).

Despite the progresses in the forecasting technologies, major challenges remain to realise the potential benefits of an EWS (Cools *et al.*, 2016). Limiting factors have been identified for Bangladesh and listed below:

- Hazard, exposure and vulnerability of the risk environment are not sufficiently evaluated through a comprehensive and shared approach among stakeholders and institutions (Fakhruddin *et al.*, 2015). In Bangladesh, flood risk is highly dynamic as river morphology, population and society constantly change. Periodical updates can provide a

good know-how of risk at the local level, thus requiring stakeholders, governmental organizations and NGOs to cooperate together with the community (i.e. Community Risk Assessment – CRA) .

- The FFWC warnings are based on WLs that do not represent the possible impacts at the local level. Instead of being distributed over large and heterogeneous areas connected to one forecast point, FFWC warnings should include



location-specific information on expected impacts and actions for different users (van den Homberg *et al.*, 2016; Rahman *et al.*, 2012; Shah *et al.*, 2012).

- Although advances in forecasting and communication technologies are improving access and timeliness of warning, there is still room to improve the design and layout of warning messages. Currently flood forecast bulletins or warning messages simply contain the forecasting station name and its expected rise or decline in WL in the coming period. Not all the exposed population is warned by forecast bulletins, especially remote locations where the existing dissemination means may not be effective.

- Warning lacks meaningful guidance information on what water levels mean and how risk mitigation actions can be supported. Improvements can be achieved by giving evidence on expected impacts and possible advices, which would require strong collaboration between FFWC, DDM and other relevant stakeholders (Shah *et al.*, 2012).

As consequences, warning thresholds might be misrepresenting the actual flood levels for which impacts occur, warnings can be imprecise for specific areas, warnings lose meaning concerning their purpose of reducing risk as risk is not specified into the message, warnings might not be implemented into actions. Improving access to IBFW services in Bangladesh could offer a mean to tackle these limitations by connecting information on the potential impacts by developing impact-based thresholds to produce more targeted and meaningful warnings that could trigger risk mitigation actions.

This research presents the potential of this approach with the aim of triggering further research and developments on IBFW in Bangladesh.

## 2    Methodology

The study follows a qualitative approach and was set up in four steps.

1. The study participants at the case study locations were identified and clustered in groups, representing the key stakeholders from different end-user groups (i.e. sectors).
2. From past events, flood impact scenarios were determined investigating flood consequences and related flood peak WLs (reference floods). Then, impact-scenario thresholds (expressed in WLs) could be determined too and impact mitigation actions for the flood-exposed groups were identified for each scenario. Impact-scenario maps were produced by transposing the reference floods WLs  at the case study locations, where detailed geographical spatial data were gathered from existing sources.
3. Colour codes were associated to each impact-scenario, thus creating impact-based and colour-coded bands. Each band represents warning ranges related to WLs by knowing impact-scenario thresholds. Colour-coded warnings were then developed: while warning messages can address main information (i.e. location, time, expected water level, etc.), the correlated colour code acts as a visual link to local guidance-information like maps, expected impacts and suggested risk mitigation actions for facing the expected consequences (Figure 1). Example of such guidance information were tested at the local level together with dissemination channels and message content required to trigger effective risk mitigation actions.



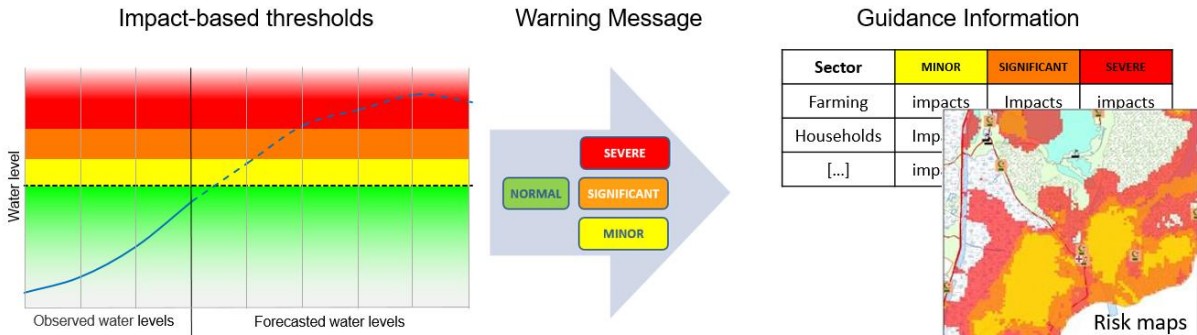

**Figure 1. Use of colour codes to link impact-based forecasting and specific warnings to trigger response at the local level in combination with guidance information.**

4.  Meetings with key authorities involved in the EWS service at both national and local level were performed in order to discuss the requirements for making IBFW an operational service in Bangladesh.

## 2.1   Community-Based Approach

A community-based approach (CBA) was chosen as data collection method for step 2 and step 3 of methodology. CBA was preferred to conventional risk assessment tools (i.e. water depth-damage curves) as water depth-damage curves at the case

study areas were not available for the considered sectors, although studies for rice crops damage assessment were performed (Kwak *et al.*, 2015). For the study's purpose, CBA had the advantage of providing site-specific local data, collected directly from the community level at the case study locations. However, data are representative for small samples, therefore highly subjective and difficult to validate. Nevertheless CBAs methods were successfully applied in previous studies, for example, to plot risk maps based on past flood events and collecting vulnerability data (Smith *et al.*, 2016; Fakhruddin, 2015;

Bahauddin and Uddin, 2012).

Data were collected during the fieldwork which took place in May 2016. Focus Group Discussions (FGDs) were used to: gather multiple responses, deepen data through dynamic discussions with participants and to test warning messages. Six FGDs, for 40 participants, were conducted. Each discussion group represented a sector of interest (i.e. farming, education and 'disaster management') in two case study areas (see Table 2). FGDs were led with the help of a checklist to pursue the

objectives of step 2 (i.e. reference floods, impact scenarios, mitigation actions) and step 3 (test of a possible impact-based warning message) as introduced in the methodology. Semi-structured interviews (SSIs) were conducted to complete step 4 (requirements for effective IBFW in Bangladesh). SSIs, with the help of a checklist, were conducted with 13 governmental officials and knowledgeable experts. For both FGDs and SSIs the answers were collected qualitatively (open answers) and quantitatively (closed answers).

Data collected from FGDs and SSIs were analysed. Local interpreters translated the response to English when needed. The answers were noted during meetings and later reported as written text. These were then analysed and used to develop the





raw data into thresholds, flood impacts, risk mitigation actions and impact-based warnings (from FGDs); institutional needs, technical arrangements and training/educational resources (from SSIs). Closed answers were related to a rating scale from 1 (low preference) to 5 (high preference) and later reported in pie charts for analysis (Likert, 1932).

## 2.2    Case study locations and participant selection (step 1)

5    This study focused on the flood-prone unions of Rajapur (Belkuchi sub-district) and Ghorjan (Chawhali sub-district), both located in Sirajganj district, Rajshahi division (Figure 2). Rajapur lies on the right bank of Jamuna River, protected for one third of its extension by a large embankment on the western side. Ghorjan mainly lies on *chars*, where no embankments are built. The closest monitoring station for which WLs are forecasted is located near Sirajganj City, the district's headquarters, roughly 15 km and 35 km upstream of Rajapur and Ghorjan respectively (Figure 2).

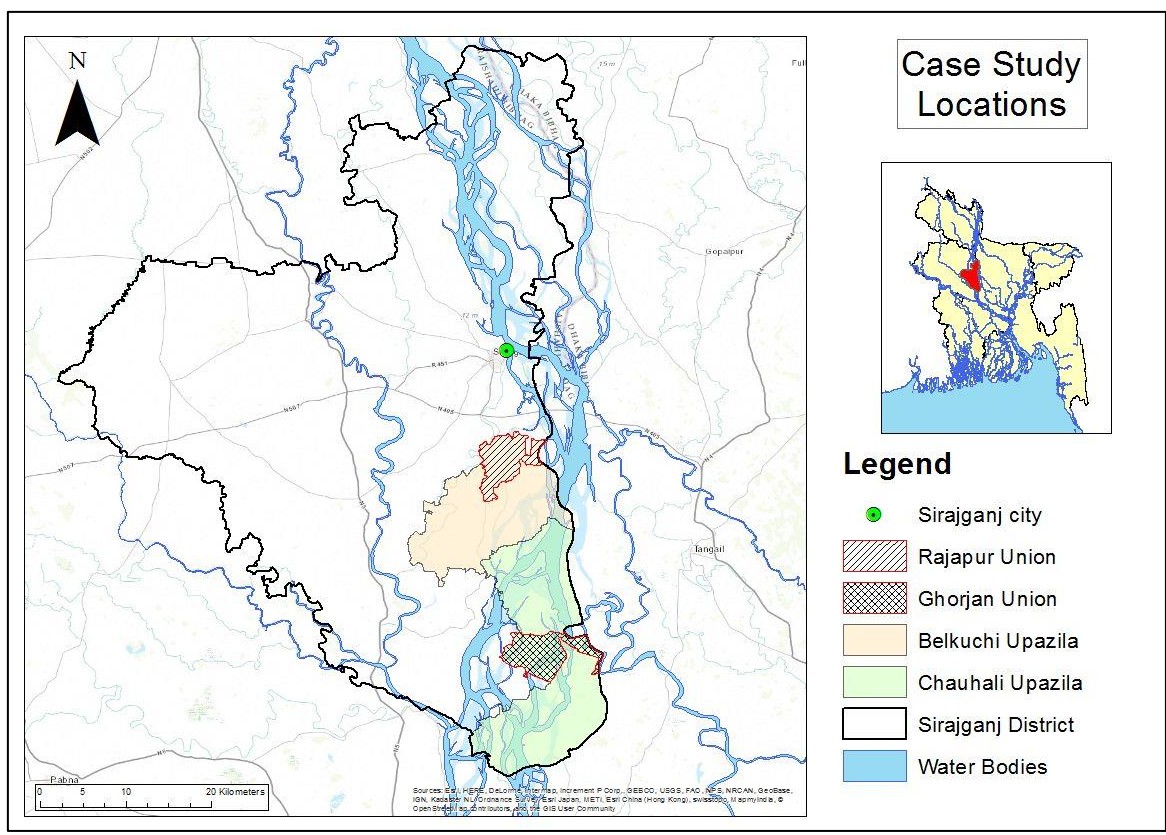

**Figure 2. The map at the top right corner shows the location of Sirajganj district. Rajapur union, located in Belkuchi upazila, and Ghorjan union, located in Chawhali upazila, are shown in the larger map. The model's WL forecasting node, coinciding with the WL monitoring station, is located on the Jamuna River in Sirajganj city identified by a green dot.**

At the case study locations, the 'farming sector' was selected as it would mostly benefit from early warnings (Fakhruddin *et al.*, 2015). The farming sectors' main activities are agriculture and domestic-scale farming (cattle, goats and vegetable gardening) which suffer from flooding every year (Fakhruddin *et al.*, 2015). Major crops grown in the study area include rice



(*boro* and *aman* qualities), sugarcane, jute, tomato, potato, onion, beans etc. Severe floods of 2004 and 2007 caused agricultural damages for more than 17,000 million BDT together (roughly 200 million USD) in Rajshahi division (Haque and Jahan 2015). The 'educational sector' was included as it's likely to be relevant for the study purposes. Students compose a large part of the local community susceptible to flood impacts as they are exposed to waterborne diseases, disruption of the

educational service and risk of drowning. Teachers can play a key role in enhancing preparedness by disseminating warnings, cooperating with NGOs and volunteers. A third sector was considered to represent the general interests of the society during emergency situation: the 'disaster management sector' including members of the local Union Disaster Management Committees (UDMCs) and the volunteers taking part in warning dissemination. Health, boat owners, rickshaw drivers and handloom industries were also identified during the SSIs as sectors that could benefit from EWS, however they

were excluded due to time limitation during the fieldwork. Table 2 summarises the groups involved in FGDs and the number of participants interviewed. The sample size limits this study, both for the number of participants and for the number of FGDs held for each sector.

**Table 2. Groups interviewed for the FGDs at the case study locations divided in sectors. In brackets the number of participants.**

| Union | Farming sector | Educational sector | Disaster Management |
|---|---|---|---|
| Rajapur | Farmers (13) | Teachers (4) | UDMC members (6) <br> Dissemination Volunteers (6) |
| Ghorjan | Farmers (5) | - | Dissemination Volunteers (6) |

### 2.3 Reference floods, impact scenarios and mitigation actions (step 2).

The data gathered was as follows:

   a) Identification of reference floods leading to minor, significant and severe impacts;
   b) Flood impacts related to the experienced events for each sector;
   c) Mitigation actions required for reducing the identified impacts.

As currently is practiced by different countries (i.e. UK MetOffice, 2017; Meteoalarm, 2017; Pagasa, 2017; Protezione

Civile, 2017; IMD, 2018), four colour-coded classes were considered in this study (according to the international standards of ISO 22324:2015). Each class represents the magnitude of flood impacts: given the normal condition as a no-impact scenario (green), minor impacts is the first level (yellow), followed by significant impacts (orange) and severe impacts (red). Therefore, neglecting the no-impact condition, three impact-scenario classes had to be determined. During FGDs at the local level, questions were asked to investigate previous flood events according to the experienced impacts. Although it was

recognized that the time of flooding influences the magnitude of the related flood impacts, an increasing linear relation with WLs was assumed for this study. By knowing the year of the events, the maximum flood peaks (reference floods) were determined on the time series of observed WLs at Sirajganj station in the period 1980-2015 (Source: FFWC). Community impact-based thresholds were identified as the average WL between the consecutive reference floods for each scenario. In case of multiple events for the same class, the one having the lowest WL (more frequent) was considered as the reference.



WLs at the case study locations were then transposed from Sirajganj station using correlation equations, assumed to be linear in this particular case, determined using measurements from staff gauges installed during a previous project at the same locations (RIMES, 2015). Only the formula for Rajapur was validated with further data and hence used in this study. Hazard maps could be plotted by subtracting the classes' reference WL from the digital elevation model (DEM) built over surveyed points (RIMES, 2015). Including vulnerability data (i.e. infrastructures, buildings, land-use, vulnerable elements, etc.; RIMES, 2015), flood risk scenarios maps linked to the appropriate colour-coded class were also plotted.

### 2.4 Impact-based warning message (step 3)

The findings gathered from step 2 were implemented in a warning message sample proposed during FGDs and SSIs. The message was kept short to reflect the space and time limitations of IVR and SMS. In these formats colour codes cannot be visualised. On the other hand, bulletins and announcements via TV/radio can more clearly indicate the colour codes. For these reasons, the example shown was prepared including colour codes also in written and spoken form (Figure 3). The example relied upon the site-specific voice call message developed under a pilot project at the case study locations (Cordaid, 2016). Note that warning does not have impact information, the use of the orange colour will convey the consequent impacts consultable from the guidance information.

"Welcome to the FFWC of BWDB. Today Friday 29th July 2016. As per the observations of 6 AM this morning Jamuna river at Sirajganj is flowing 10 cm below Danger Level. According to the latest flood forecast water may rise 29 centimetres in Rajapur union, in next 5 days. **Yellow code. Flood may occur in 4 days in Rajapur union**"

"Welcome to the FFWC of BWDB. Today Friday 29th July 2016. As per the observations of 6 AM this morning Jamuna river at Sirajganj is flowing 10 cm below Danger Level. According to the latest flood forecast water may rise 30 centimetres in Ghorjan union, in next 5 days. **Orange code. Flood may occur in 4 days in Ghorjan union**"

**Figure 3. Warning messages shown during FGDs and SSIs as an example of a possible impact-based warning. The existing voice call warning was improved by adding the bolded text and shading the text with the colour code.**

The information gathered for testing impact-based warning messages was as follows:

a)  the sequence of colours for expressing impact severity comparing the best practices model (green, yellow, orange and red) and the practice currently used at FFWC (green, yellow, magenta, red);
b)  test for understanding the acceptance of including colour codes into warning;
c)  the most appropriate dissemination channels;
d)  the overall grade of appreciation for delivering impact-based information (Figure 1).

### 2.5 Operational requirements for IBFW service in Bangladesh (Step 4)

The mechanisms through which the different institutions could distribute responsibilities to generate an IBFW was explored. An IBFW service should be performed on a daily basis. Such tasks would include translating the WLs forecasted by FFWC




at the forecasting stations into a more location-specific, impact-based and colour-coded warning. SSIs' interviewees were asked what organization would be the most suitable for fulfilling this task and what geographical size IBFWs should be prepared for. More generally the investigation aimed to identify the requirements in terms of:

 a) institutional requirements (i.e. responsibilities and operational framework);
 b) technical requirements (i.e. forecasting network, WL gauges);
 c) capacity building requirements (i.e. workforce, volunteering activity, awareness and drills).

## 3    Results and discussion

### 3.1    Reference floods, impact scenarios and mitigation actions (step 2)

Table 3 shows the past flood events identified as reference floods (section 2.3 – objective a)).

10  **Table 3. Classification of reference floods in three scenarios based on the perceived impacts concerning the sector of interest of the interviewees as experienced in previous flood events. In bold, the event chosen as reference flood (lower WL) in case of multiple answers given for the same scenario.**

| Discussion group | Location | Perceived impact severity (year) | | |
|---|---|---|---|---|
| | | Minor consequences | Significant consequences | Severe consequences |
| Farmers | Rajapur | 2014 | 2015 | **1988**, 2007 |
| Teacher | Rajapur | 2014 | 2015 | 2007 |
| UDMC | Rajapur | 2014 | 2015 | **1998**, 2007 |
| Dissemination volunteers | Rajapur | 2014 | 2015 | 2007 |
| Dissemination volunteers | Ghorjan | 2013 | 2014 | 2007 |
| Farmers | Ghorjan | 2013 | **2014**, 2015 | 2007 |

Looking at Table 3, there is a good match within the same geographical area across sectors. However, 'minor' and
15  'significant' between Rajapur and Ghorjan were identified differently. This aspect highlights the importance of having more geographically specific forecasting as the hydraulic conditions may differ significantly along the river, leading to imprecise warnings issued according to wrong thresholds. In Rajapur three events were identified as belonging to the 'severe' scenario. On the other hand, only the 2007 flood was mentioned as severe event from Ghorjan's FGDs. It might be either that the previous events was not experienced by Ghorjan inhabitants or people remembered only the most recent. Perhaps, that *char*
20  was not formed yet (Paul and Islam, 2015).

Table 4 shows the reference floods' WL for each scenario, transposed from Sirajganj station to Rajapur union using existing correlation formulas available (described in Step 2 of Methodology) for two gauges placed within the union boundary. These were used to plot hazard maps (Figure 4) and thus delimiting the inundated areas.


**Table 4. The flood peak values observed at Sirajganj were transposed at Rajapur union using existing correlation formulas available for two gauges at Randhunibari and Thakurpara placed within the union boundary of Rajapur. The average of them was used to produce hazard maps at Rajapur Union. WLs are reported with respect to the Public Works Datum (PWD) of Bangladesh, which is 0.46 m below the Mean Sea Level (MSL).**

| Impact scenario for Rajapur | Year | WL at Sirajganj (mPWD) | Return period (Years) | WL at Randhunibari (mPWD) | WL at Thakurpara (mPWD) | Average WL (mPWD) |
|---|---|---|---|---|---|---|
| Minor | 2014 | 13.79 | 1.52 | 13.00 | 12.74 | 12.87 |
| Significant | 2015 | 14.13 | 2.46 | 13.33 | 13.07 | 13.20 |
| Severe | 1998 | 14.76 | 12.14 | 13.95 | 13.69 | 13.82 |

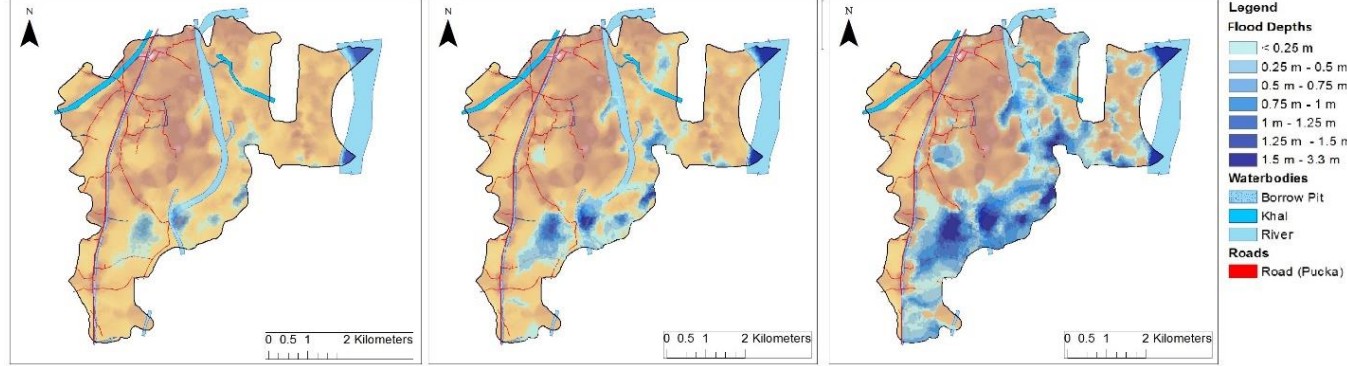

**Figure 4. Hazard maps plotted for Rajapur union. From the left: minor event (2014), significant event (2015) and severe event (1988).**

Figure 5 represents the peak WLs – recorded at Sirajganj station - of the events mentioned in Table 3 and the sectorwise

10 impact-based warning bands (impact scenarios) determined with the CBA, plotted for both unions. This result shows that CBA findings can be different in different locations. The differences may also be attributed to small sample sizes, and we hope in the future the presented methodology will be explored further with larger sample sizes at diverse locations. The colour bands identify the colour-coded warning representing both impacts and the flood severity (hazard) leading to these. Colour codes can then link to guidance information for each sector (e.g. maps, expected impacts and suggested risk

15 mitigation actions).




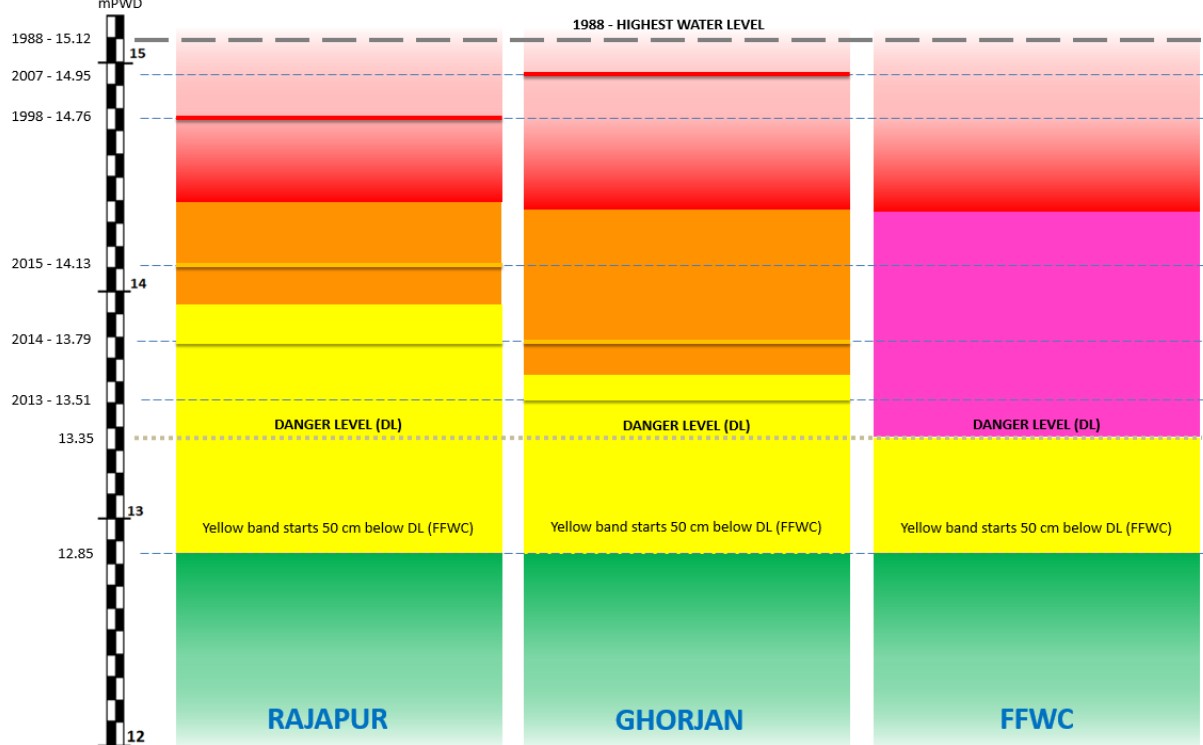

**Figure 5. Site-specific warning thresholds determined with the CBA for each union with respect to Sirajganj station. On the right, the colour bands as per at FFWC for Sirajganj station. On the left, the stage reports the classes' reference WLs and the year they occurred. Thresholds between bands were obtained as average between two reference WLs (see table 4). The yellow warning band starts 50 cm before the danger level, as per the current application at FFWC.**

For each scenarios, experienced impacts and necessary risk mitigation actions were collected through FGDs and reported in Table 5 and Table 6. Tables like these might be important for managing risk and the post-event aftermaths, additionally they provide information available for increasing preparedness before the event.

**Table 5. Impacts experienced for previous flood events by the sectors involved in the research in Rajapur community. WL is referred at Sirajganj station in mPWD.**

| Impacts for Rajapur union's community | | | |
|---|---|---|---|
| **Sector** | **Minor events** <br> 12.85 < WL< 13.96 <br> Duration: 1-2 weeks | **Significant events** <br> 13.96 < WL < 14.45 <br> Duration: 3-4 weeks | **Severe events** <br> WL > 14.45 <br> Duration: >5 weeks |
| Farming sector | Crop damages <10%. | Crop damages 60%-80%; <br> Loss of livestock. | Crop damages 80%-100%; <br> Diffuse loss of livestock. |
| Educational sector | School closure (day/s) | School closure (week/s); | School closure (several weeks). |
| Disaster Management sector | Households inundated; <br> Minor electricity cut-off; <br> Few roads inundated. | Households inundated; <br> Electricity cut-off; <br> Roads inundated; | Households inundated; <br> Electricity cut-off; <br> Many roads inundated; |



| | | Lack of drinking water; Some cases of waterborne diseases; Lack of dry wood; Daily labours abruption; Snake bites; Soil erosion. | Lack of drinking water; Diffuse cases of waterborne diseases; Food scarcity; Lack of dry wood; Daily labours abruption; Snake bites; Soil erosion. |
|---|---|---|---|

**Table 6. Actions that participants would take for reducing risk in their sector, based on previous flood events in Rajapur union. WL is referred at Sirajganj station in mPWD.**

| Risk mitigation actions for Rajapur union's community | | | |
|---|---|---|---|
| **Sector** | **Minor events** 12.85 < WL< 13.95 Duration: 1-2 weeks | **Significant events** 13.95 < WL < 14.45 Duration: 3-4 weeks | **Severe events** WL > 14.45 Duration: >5 weeks |
| Farming sector | Preventive harvesting; Protective embankments; Animal displacement; Pond netting. | Preventive harvesting; Protective embankments; Animal displacement; Pond netting; Animal vaccinations. | Preventive harvesting; Protective embankments; Animal displacement; Pond netting; Animal vaccinations. |
| Educational sector | Food and water storage; Dry clothes for children. | Food and water storage; Dry clothes for children; Class relocation. | Food and water storage; Dry clothes for children; Class relocation. |
| Disaster Management sector | Items relocation; Roads reinforcement. | Items relocation; Roads reinforcement; Evacuation to relatives; Saving of money, food, water, cooking stoves, fuel for generators. | Items relocation; Roads reinforcement; Evacuation to relatives; Saving of money, food, water, cooking stoves, fuel for generators. |

5  Impacts determined for the 'farming sector' confirm the findings of previous studies (Fakhruddin *et al.*, 2015; Bhuiyan and Al Baky, 2014; Dewan 2014). However, farmers in both unions acknowledged inundation extent and timing of flood as important information for their sector for defining impacts and taking risk mitigation actions (Fakhruddin *et al.* 2015). In some cases, same impacts were mentioned for more sectors (i.e. education and disaster management). It suggests that flood impacts can be experienced sector wise, however, it is acknowledged that results could differ if a larger sample was used.

10  Furthermore, same impacts might occur for all the severity classes, however negative consequences are directly linked to both inundation extent and flood duration. In the direction of IBFW, researches shows that the local community in Bangladesh do implement risk mitigation actions based on local experience as long as location-specific information for the oncoming hazard  are issued (Cumiskey *et al.*, 2015; Fakhruddin *et al.*, 2015; Maidl and Buchecker, 2015; Oktari *et al.*, 2014; Shah *et al.*, 2012).




The CBA has limitations. First, the FGD participants were selected among the members involved in a project working in the same locations (Cordaid, 2016). However, Rahman *et al.* (2012) showed that flood risk awareness is remarkably high also for individuals living in remote communities but not participating in similar programmes. Second, the CBA in this study was performed over a small number of interviewees, all belonging to the same village, where both flood categories and impacts

may not be representative for other settlements in the same union. Other impact-assessment methods, like water depth-damage curves (DDCs), are determined over adequate sample sizes and hence representative for larger communities. However, this strength might turns into weakness as specific needs and differences could be underestimated. Furthermore, DDCs are usually determined for few land uses (i.e. agriculture and settlements), not considering other stakeholders thus excluded from impact assessment. On the other hand, CBA ensures that the local needs are considered and can be

representative for other sectors of interest. For the purposes of this study, flood categories could be clearly determined with a CBA and hence used to tailor IBFW thresholds. Although the data collected through the CBA proved to be useful for the purpose of this research, future applications would require more in-depth data collection to validate results. This may not be feasible in many applications presenting a challenge for using the approach.

### 3.2 Impact-based warning message (step 3)

The following list reports the results gathered according to the points presented in step 3 of the methodology.

    a) The entire FGDs sample (40 individuals) expressed the green-yellow-orange-red colour code sequence as more intuitive than the sequence currently shown on FFWC website. Participants demonstrated interest in colour-coded warnings as it is easier to interpret flood severity. Bulletins, currently not reporting colours codes, can benefit from the linkage of severity classes and colour codes.

b) The example reported in Figure 3 was shown to a sample of 48 individuals (FGDs and SSIs). Eighty-three percent of them strongly agreed with the inclusion of colour codes into the warning message as it is a more intuitive framing of risk and consequent risk mitigation actions, 11% acknowledged the benefits, while 4% and 2% respectively remained neutral and did not agree. This result shows that colour codes can already provide an information for taking risk mitigation actions without the need of interpreting numbers (i.e. WLs) (Fakhruddin *et*

*al.*, 2015). However some end users might favour the existing warning message (i.e. voice call without implementation of colour codes) as colours can be misunderstood.

    c) Guidance information can be shown either online on web-based applications or on printouts/boards made available at key locations at the local level (i.e. governmental buildings, schools, bazaars, mosques, etc.). Volunteer based warning dissemination was agreed to be an effective solution to reach the very rural areas where current

technologies are either not available or people's ability to understand warning messages is limited. During FGDs it was recognised that multiple reinforcing communication mediums are needed to reach everyone at risk (Oktari *et al.*, 2014). Despite the poor conditions of the local communities, the possibility of using mobile phones for





dissemination was widely accepted (Cumiskey *et al.,* 2015). However, respondents indicated that remote areas could experience shortage of electricity for long periods, making those devices less reliable.

d)  Seventy-six percent of the sample, composed of 49 interviewees (FGDs and SSIs), strongly agreed that linking colour-coded warnings with guidance information at the local level would benefit the exposed population towards specific risk mitigation actions. Twenty-two percent acknowledged the utility of it, while only 2% disagreed. Interviewees mentioned institutional buildings, UDCs, bazaars, schools and mosques as locations where to consult guidance information materials. Across the sectors, many stated that the maps' resolution should be focused at the smallest scale, namely villages. Having this level of detail would permit the inclusions of vulnerable elements and critical infrastructure, useful for coordinating relief efforts and used as tool for fostering agencies collaboration (van den Homberg *et al.*, 2016; Dewan *et al.*, 2014; Scolobig *et al.*, 2012). Although this guidance information can be provided, further efforts are required to train and educate end users on how to interpret warning messages, for example with documentaries, drills and media held in the local language (Cumiskey *et al.*, 2015; Scolobig *et al.*, 2012).

It is recognised that these results are very positive and realistically this could vary significantly if tested during a real time situation. Further testing and applied research is needed to define the most suitable IBFW.

### 3.3  Requirements for IBFW service in Bangladesh (Step 4)

Implementing IBFW in Bangladesh will not happen overnight. It requires a long process of upgrading technology, building capacity and enabling both institutional and policy change. We propose some guidance on possible steps that could be taken in the short and long term to move towards IBFW in Bangladesh.

a)  Regarding the institutional requirements, in the short-term, the FFWC can take simple steps to integrate area-wise colour coded warnings in the current FFWC website and to set up an interagency team to decide on the guidance information at the national/ district level, whilst continuing to improve the FFWC water level gauge network and modelling capabilities. Such a team would consist of the critical agencies including FFWC, DDM, BMD and if possible some sector specific agencies e.g. health, agriculture. In this case, all the forecasting would be done at the national level but support could be provided on how to interpret these colour coded warnings at local level to provide guidance. Experimentation with probabilistic forecasting and communication and local pilot projects should continue especially research on the behavioural response of recipients to impact/ colour coded messages. In the long run, it is suggested to develop a neutral interfacing agency that joins these institutions both at national level (as above) and at the sub-district level to connect the Upazila government office with the district level BWDB office and local government institutions and other locally relevant people (e.g. UDC entrepreneurs, local champions). The sub-district level was found to be a more appropriate level for performing this task on a daily basis given their manpower and potential access to resources. Such an agency would be responsible to translate the forecasts from





the national level to the local context and provide meaningful guidance information for different colour coded warnings. These warnings can then be further communicated to local volunteers via mobile services, accessible via the UDC entrepreneurs via internet and maps, and shared via mobile and in-person to individuals. This arrangement would require significant investment and resource allocations for skills and technology development within existing agencies, newly developed agencies and local volunteers, to monitor water levels, produce localised forecasts, and communicate them. Furthermore, legislative changes in the Standing Orders on Disasters are required to allocate new roles and responsibilities for IBFW and engrave the institutional changes.

b) On the technical requirement' side, specific warnings can be prepared only if WLs can be predicted at smaller geographical areas, namely at the union level. It is now possible in the short term by correlating WLs of the forecasting stations with hydrometric stages placed at the locations where specific warnings have to be issued to. In the long term, a dense forecasting network should be developed by FFWC/BWDB, ensuring a good flood forecasting coverage as identified by Oktari *et al.* (2014) and Shah *et al.* (2012). High resolution DEMs are needed for mapping (forecasted) inundation areas (Kwak *et al.*, 2015; Fakhruddin *et al.*, 2015; Koks *et al.* 2015; Bhuiyan and Al Baky, 2014). However, the rapid morphological alterations of the main watercourses might create a challenge requiring periodical updates. Similarly, vulnerability data should be periodically surveyed, mapped and shared for coordinating emergency plans at the local level (Dewan *et al.*, 2014; Scolobig *et al.*, 2012).

c) For capacity building requirements, workforce needs expertise of the forecasting system and technical tools for warning preparation (i.e. digital services, maps, etc.). Volunteering activity requires training/drills and coordination with governmental and non-governmental organizations (Cumiskey *et al.*, 2015; Scolobig *et al.*, 2012). Volunteering activity showed to be an effective means for message delivery, IBFW can trigger its inclusion as institutional practice as per other hazards in Bangladesh (i.e. storms). Education campaigns need to be planned and periodically reiterated, ensuring a high proactivity of the community level, thus creating a direct communication channel for being updated with end users' needs (i.e. for tuning IBFW thresholds in future) (Cumiskey *et al.*, 2015; Fakhruddin *et al.*, 2015).



Figure 6 schematizes the approach proposed for delivering site-specific impact-based warnings in the short and long term based on the considerations drawn above.

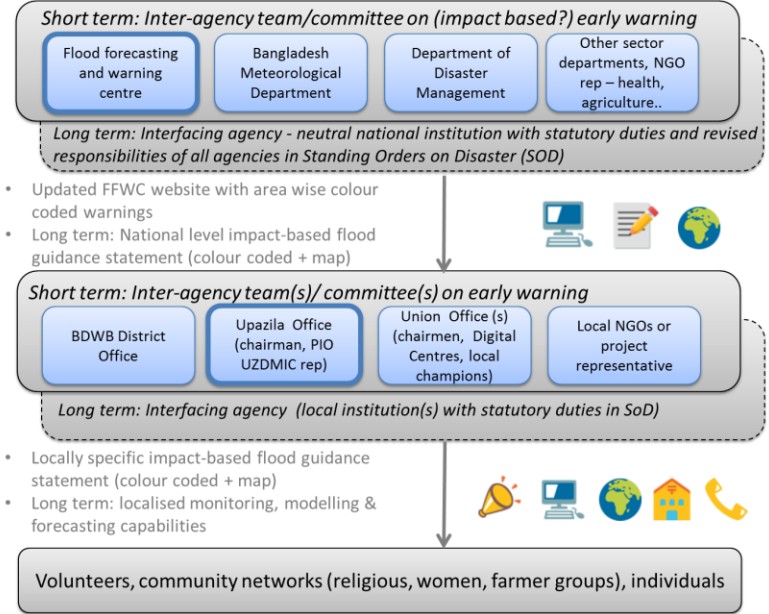

**Figure 6. Proposed approach for delivering site-specific impact-based warnings at the local level. The framework shows both short-term and long-term solutions proposed in this study for delivering more tailored and understandable impact-based warning to different groups of users. Bolded frames represent the main actors identified for these tasks in the short term.**

## 4    Conclusions

IBFW services require site-specific forecasting and warnings issued in an understandable way able to trigger risk mitigation actions by informing end users about the expected impacts. This research presented some challenges of IBFW and proposed an approach for implementation in Bangladesh. We investigated the use of colour-codes to relate flood warnings with expected impacts, impact  maps and suggested risk mitigation actions, grouped in three scenarios. Each scenario represents a range of water levels identified by a colour band that might lead to similar impacts as experienced in the past. Impact-based warnings can be produced for those forecasted water levels falling into the identified scenarios. Overall, by improving the existing pilot projects outcomes carried out at the case study locations, the colour coded impact-based warnings were found to be an easy and understandable way to link water level forecasts to the necessary risk mitigation actions in comparison to the existing FFWC bulletin.  However, further applied research is needed to validate these findings under real-time conditions and to properly investigate the linkage of colour codes with guidance information. Furthermore, studies have to be performed to identify the appropriate medium for communicating such messages according to the behavioural responses to these.  IBFW has huge potential in Bangladesh but its integration requires significant institutional changes, such as an inter-facing agency (long term) or team (short term), adjusted policy frameworks (standing orders on disasters), and new





resource allocations for skills development and technological innovation from national to local levels. More specifically, the spatial domain of the forecasting model, the density of WL gauges and their correlations, and high resolution risk assessments, need to be improved. Resources are required to develop technical skills, increase manpower and coordinate volunteering activities at the local level. Future studies can deepen these findings by investigating up-scaled applications by

increasing samples for different sectors, comparing the results with other impact assessment methods (i.e. water depth-damage curves) and doing deeper analysis on the behavioural responses to IBFW warnings in real-time at the local level. Furthermore, research needs to be carried out for correlating the dynamic nature of varying flood impacts in space and time on a large scale that can be still representative of the local context.

## 5   Acknowledgments

The main author wants to thank Deltares for funding the fieldwork and the European Commission for partially funding the European Joint Master Programme in Flood Risk Management. For technical help and support in Bangladesh: Mr. Amirul Hossain (FFWC/BWDB), Mr. Netai Sarker (DDM), Mrs. Wahida Bashar Ahmed (Cordaid), Mr. Rafiqul Islam Khan and Mr. Mehedi Hasan (Manab Mukti Sangshta).

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
