# Peer review of "Towards impact-based flood forecasting and warning in Bangladesh: a case study at the local level in Sirajganj district"

_Natural Hazards and Earth System Sciences, 2018_

## Referee Comment (RC1) · M. Zappa (Referee) · 6 Apr 2018

Dear authors,

This is surely a nice piece of work for the specific community addressed. I fear, that the relevance outside this community is at this stage quite limited.

Manuscripts reads more like an NGO report and less like a scientific paper, but I have to admit that I am not used to read/write manuscripts with much text and few facts as usually done in the risk-management community.

Further remarks:
(Too) Large use of abbreviations do not ease the reading.

See the annotations in the manuscript for further comments.

Best regards

Massimiliano Zappa
6.4.2018

Please also note the supplement to this comment:
https://www.nat-hazards-earth-syst-sci-discuss.net/nhess-2018-26/nhess-2018-26-RC1-supplement.pdf

**Supplement:**

[revised manuscript text omitted]

---

## Author Comment (AC1) · 16 May 2018

The authors would like to thank the reviewer for providing very relevant feedback to our work. We think that the comments will help us to improve the manuscript and increase the chance of its publication. In the following we are providing our replies to the comments.

Comment # 1: Page 1, line 1: On the scientific relevance of the work beyond the specific community addressed in the research. Response # 1: As a response to your general comment regarding the scientific relevance of our work, we would like to remark that Bangladesh is a country prone to large scale fluvial floods and those who are af-

fected, may be from different locations, are often belonging to the same social context. In fact, although communities differ, similar local structures can be found elsewhere (i.e. farmers cultivating agricultural field in low lying areas and/or domestic farming, households, small-scale industries, etc.). Thus, the main point of our case study is indeed to stand key elements that can be tested and deepened through further research: impact-based warning thresholds and an approach for delivering tailored flood warnings. We sincerely believe that this work can address the scientific community towards innovative and feasible impact based forecasting techniques for developing countries, which is spearheaded by the World Meteorological Organization's input.

Comment # 2: Page 1, line 1: Manuscripts reads more like an NGO report and less like a scientific paper. Response # 2: We agree that the manuscript is verbose. This is mainly due to the topic that it addresses. Impact based flood warning is a new topic and its usage in countries like Bangladesh is limited. Therefore, there is a need to portray the state-of-the-art of the flood warnings and its limitation. This part will have to be descriptive. In the revised manuscript we will explore to present the manuscript in less descriptive way, including adding more figures and tables.

Comment # 3: Page 1, line 1: Large use of abbreviations do not ease the reading. Response # 3: We agree that too many abbreviations have been used and they hinder easy reading. We are avoiding to provide a list of them as that may not help in easy browsing through the text. We will work on the revised manuscript to remove some of the details and in further abstracting information so that the need of abbreviated names of organisations and procedures will be less required. This response will also (partly) address the main comment of the reviewer about the scientific relevance of the manuscript beyond the specific community addressed in the study.

Comment #4: Page 1, line 26: "First insight" on a "case study "is not really a great selling argument for the manuscript Response #4: The phrase "first insight" does not refer to the case study but towards the critical analysis of possible impact based forecasting in Bangladesh illustrated with a case study.

Comment # 5: Page 2, line 10: Is all this part of the (UNISDR, 2006) citation? Response # 5: Yes it is, we will revise the manuscript for clarifying this citation according to the newer issue of 2018 https://ane4bf-datap1.s3-eu-west-1.amazonaws.com/wmocms/s3fs-public/ckeditor/files/Multi-hazard_Early_Warning_Systems_A_Checklist.pdf?fVgoQYM7LhPb3oR0V97j2.Qkjs3Wc5Rq

Comment # 6: Page 2, line 19. "I miss here examples of IBFW implemented in other areas". Response # 6: The examples are briefly provided in the existing manuscript. We request the attention to the existing text on page 2, line21: "Although recently introduced, best practices can be found in national meteorological services, like the UK Meteorological Office (UK MetOffice, 2017) and the United States National Weather Service (US NWS, 2017), as well as in international programs leaded by WMO through dedicated workshops (WMO, 2017)."

Comment #7: Page 2, line 26: Suggested references: You might find also information in Ronco, P., Gallina, V., Torresan, S., Zabeo, A., Semenzin, E., Critto, A., and Marcomini, A.: The KULTURisk Regional Risk Assessment methodology for water-related natural hazards – Part 1: Physical–environmental assessment, Hydrol. Earth Syst. Sci., 18, 5399-5414, https://doi.org/10.5194/hess-18-5399-2014, 2014.

Ronco, P., Bullo, M., Torresan, S., Critto, A., Olschewski, R., Zappa, M., and Marcomini, A.: KULTURisk regional risk assessment methodology for water-related natural hazards – Part 2: Application to the Zurich case study, Hydrol. Earth Syst. Sci., 19, 1561-1576, https://doi.org/10.5194/hess-19-1561-2015, 2015. Response # 7: We will go through the suggested references and will add citations. We are thankful to the reviewer for the valuable suggestion.

Comment #8. Page 4, line 3: This does not belong in the introduction. Response #8: We thought of providing the text on state-of-the-art of flood forecasting and warning in the introduction section. Based on the reviewer's suggestion we will consider placing it as an introductory text in Section 2.2 (Case study locations and participant selection).

Comment # 9: Page 5, line 19: "Are you the first using this first step approach? How did other researcher plan the setup of a IBFW". Response # 9: As mentioned, IBFW is a new initiative without many applications. Some applications are mentioned in the manuscript. However, we have indeed missed mentioning them in the beginning of Section 2. We will refer to the existing success stories in the revised manuscript.

Comment #10: Page 7, Fig 2: Is there any "char" that could be highlighted in Figure 2? Response #10:Figure 2 will be updated to include a "char".

Comment #11: Page 9, line from 1 to 6: "Very vague and qualitative description on how warning levels have been set. This can be surely improved". Response #11: We will work on to bring more clarity to this section.

Comment #12: Page 9, line 17: Minor: this Orange is to me much closer to red than to yellow. Response #12: Agreed, we will work on Figure 3 to change the colour.

Comment #13: Page 10, line 10: The reader has to believe you here. Anyway, it seems to reign concordance with respect to the perceived severity of flood impacts. Response #13: Participants at the focus group discussions answered spontaneously and in most of the cases all agreed on a single response. In case there was less concordance on a single event they perceived as representative for a particular scenario, multiple events were then considered and inserted in table 3. We decided to choose the one with the lower recorded water level.

Comment #14: Page 11, line 5: "Is there any way to put uncertainty bands in these quite precise numbers?" Response #14: We agree on putting a confidence interval. It can be done by adding the upper and minor values to the calculated one according the standard deviation of the correlation formula.

Comment #15: Page 13, line 4: "How can one justify that vaccinations are needed only starting from ORANGE level, while all other mitigation measures are useful also for minor events?" Response #15: The main point in this case is not only related to flood

levels but also to flood duration. Indeed flood impacts for the significant events (orange code) are related to flood duration of 3-4 weeks. With such a long duration waterborne diseases spread among domestic animals. We will add this explanation in the revised manuscript.

Comment # 16: Page 14 – line from 1 to 14. "I have here the impression, that the authors recognize and acknowledge that this study is possibly out too early, and that higher number and more independent samples might finally lead to different outcomes with respect to the IBFW". Response # 16: Our main goal is not to demonstrate that the approach studied in this research is a solution that can be applied anywhere. Indeed the data was limited, however, the study demonstrated the usefulness of the approach. The authors have presented a critical analysis and have pointed out the limitations, which point out that for drawing more general conclusions about the approach a larger data sample representing larger area over several flood events will be required. This does not however, contradict presenting a proof of concept with a limited dataset.

Comment #17: Page 14, line 20: It would be more fair to stray with the "individuals" here and do not calculate the percentages. Response #17: We agree with this observation and we will revise the manuscript.

Comment #18: Page 15, line 3: It would be more fair to stray with the "individuals" here and do not calculate the percentages. Response #18: We agree with this observation and we will revise the manuscript.

Comment #19: Page 15, line 26: And suddenly you speak about "probabilistic fore-casting", without having introduced it before. See fore example: Bruen, M.; Krahe, P.; Zappa, M.; Olsson, J.; Vehvilainen, B.; Kok, K.; Daamen, K., 2010: Visualizing flood forecasting uncertainty: some current European EPS platforms - COST731 working group 3. Atmospheric Science Letters, 11, 2: 92-99. doi: 10.1002/asl.258 Response #19: We agree with this observation and we will revise the manuscript by being more precise according to "probabilistic forecasting" already in the introduction.

Comment #20: Page 17, line 4: Summarizing: acceptance of generally accepted colour codes could be observed among the local stakeholders. Response #20: Indeed this is the main conclusion this study wants to lead to. In parallel with this outcome, we wanted to highlight the benefits, the limitations and the challenges for up-scaling the proposed methodology.

We thank the reviewer for providing valuable comments. These comments will certainly help in improving the manuscript.
* * *

---

## Referee Comment (RC2) · Anonymous Referee #2 · 11 Jun 2018

Review of "Towards impact-based flood forecasting and warning in Bangladesh: a case study at the local level in Sirajganj district" by Sai et al. The paper describes a case study carried out to test different ways of relaying information on floods to stake-holders. The idea was to use colour-coding based on impact levels designed and decided by the sector experts themselves. Although I find the study very interesting and well-timed, I am reluctant to call this a research paper. It reads more like a preliminary report or an opinion paper. I am in favour of publishing the paper, but struggle to see where it could fit. I therefore restrain to make a recommendation, but leave that in the hands of the editor. Scientific contribution The major limitation of the study is that it is setup as a social science study, but without the social science carried out

within it. It would have been very useful to better understand how and why the actors understand or do not understand/trust/use forecasts in impacts. This would have made an entirely different paper which would have contained analysis and conclusions and a way forward to improve the forecasting. As it stands now, it is more an opinion piece, or a preliminary study on how to sue the forecasts. This is also interesting, the but scientific analysis is missing. I would therefore either reject the paper on the grounds of not being scientific, or rather transfer it to an opinion paper and allow the authors to be even more bold in discussing the pros and cons as well as outlinng future research in more detail. Presentation The paper is generally well-written, but the figures needs to be improved. if this turns into an opinion paper, I would suggest to restructure the paper as well, to follow the line of reasoning of a discussion paper. Minor comments P1, L24. Change to "short-term" and "long-term" P2, L10. Please put proper references to this statement "A complete and effective EWS comprises of four inter-related elements: a) risk knowledge, b) monitoring and warning service, c) dissemination and communication and d) 10 response capability." P2. L19. I am not convinced by this: "The term "impact-based  aims to translate the hydro- meteorological forecast by shifting the paradigm towards end users, which is forecasting the expected consequences of hazards for different sectors of interest.". Impact-based forecasts are useful at all levels, not only end-users. Also, forecasters should be helped by them in order to categorize the hazards in different risk levels. P4, L8-24. This description is very detailed, and I suggest to shorten it P5. L22-26. I am not sure I fully understand the method of creating the impact-based forecasts. Were these done individually for each point?

---

## Author Comment (AC2) · 23 Jul 2018

The authors would like to thank the reviewer for providing very relevant feedback to our work. We think that the comments will help us to improve the manuscript and increase the chance of its publication. In the following we are providing our replies to the comments.

Comment #1: "Although I find the study very interesting and well-timed, I am reluctant to call this a research paper. It reads more like a preliminary report or an opinion paper". Response #1: Thank you for this comment, the authors acknowledge the limited data collection, however, it does include qualitative data collection at the community

level, develops a framework to assess impact based forecasting and provides practical recommendations for exploring this topic further. An impact-based approach to warning communication is strongly advocated for by World Meteorological Organisation but there is a shortage of studies outlining how this can translate in practice at the community level. The authors acknowledge the limitation of the study but strongly believe that it does contribute to advancing knowledge on impact-based forecasting and warning, which will provide a useful stepping stone for further qualitative and qualitative based research studies on this topic.

Comment # 2: "The major limitation of the study is that it is setup as a social science study, but without the social science carried out within it. It would have been very useful to better understand how and why the actors understand or do not understand/trust/use forecasts in impacts. This would have made an entirely different paper which would have contained analysis and conclusions and a way forward to improve the forecasting". Response #2: Thank you for this comment. We recognise that the participants sample size limits this study but we do believe it still merits a social science study. Focus Group Discussions with 40 participants and semi-structured interviews (SSI) with 13 participants were conducted as part of a qualitative study. Furthermore, this study is built upon substantial previous research conducted in the same communities in Sirajgang – Rajapur and Ghorjan Union. These previous studies did specifically focus on the understanding, trust and usefulness of forecasts as part of a pilot project which delivered a more localised forecasts at union level via voice based SMS - see report Cumiskey et al. (2015) . Furthermore, the national level flood forecasting and warning centre (FFWC) does have national level colour coded warnings but these are not impact-based, nor are these colours used to communicate warnings at local level. Therefore, the objective of this current research was to further understand how the communities could relate to more impact-based warning thresholds and colour-coded information which is currently not being used as an approach for warning communication in Bangladesh but is being advocated for by the World Meteorological Organisation. This current research is driven from a lot of knowledge working with communities to improve flood warning communication at the local level in Siragjang. The background studies behind this research can be further clarified in the manuscript text to justify the focus. Reference: Cumiskey L., Hakvoort H., Altamirano M. Mobile Services for Flood Early Warning in Bangladesh: Final Report, Deltares, available at: https://www.deltares.nl/app/uploads/2015/11/Deltares-Mobile-Services-for-Early-Warning-in-Bangladesh-Final-Report_web.pdf, 2015.

Comment # 3: "As it stands now, it is more an opinion piece, or a preliminary study on how to use the forecasts. This is also interesting, the but scientific analysis is missing". Response #3: Thank you for this comment, as explained in Comment 1, this manuscript is a study exploring a subject which still requires further testing and application. Although not primarily focused on the forecasting system, scientific analysis was performed for studying flood extent and risk mapping. This was done by correlating the registered water levels (peaks of the reference floods) at the forecasting station to the water levels transposed at the case study locations. We do understand that limitations might affect the research quality, however the presentation is based on real data and good experience working with the local community on previous research on flood warning communication. The paper is just a step towards understanding how impact-based forecasting can be applied in practice and requires more in-depth studies to further increase scientific knowledge on this topic. We strongly consider this research to be more than an opinion piece or preliminary study and hope that it can encourage more much needed studies on this topic.

Comment # 4: "The paper is generally well-written, but the figures needs to be improved". Response #4: We are thankful to the reviewer for this positive comment. We will update the figures.

Comment # 5: "P1, L24 Change to short-term and long-term". Response #5: We agree with this observation and we will revise the manuscript.

Comment # 6: "P2, L10. Please put proper references to this statement: A complete and effective EWS comprises of four inter-related elements: a) risk knowledge, b) monitoring and warning service, c) dissemination and communication and d) 10 response capability." Response #6: We agree with this observation and we will revise the manuscript by adding the updated reference (Multi-hazard Early Warning Systems: A Checklist, WMO, 2018).

Comment # 7: "P2. L19. I am not convinced by this: "The term "impact-based aims to translate the hydro- meteorological forecast by shifting the paradigm towards end users, which is forecasting the expected consequences of hazards for different sectors of interest." Response #7: This statement was rephrased from the Guidelines issued by WMO (2015). Therefore the missing reference will be added.

Comment # 8: "P4, L8-24. This description is very detailed, and I suggest to shorten it". Response #8: We agree with this observation and we will revise the manuscript by adding references instead.

Comment # 9: "P5. L22-26. I am not sure I fully understand the method of creating the impact-based forecasts. Were these done individually for each point?". Response #9: According to literature, impact-based forecasts are mainly available for developed countries, and in these cases impacts could be assessed thanks to methods relying on vulnerability and hazard data, if available. Due to data limitations, we wanted to investigate a different approach like explained in the manuscript and here recalled. First, it was decided to define three forecast/warning scenarios (yellow, orange, red). Then the flood scenarios were investigated for each sector through focus group discussions by asking participants to identify previous events that led to minor (yellow), significant (orange) and severe impacts (red). By knowing the events it was possible to estimate the water levels at the case study location. Thus water level ranges for each scenario (yellow, orange and red) were determined. Thus, forecasted water levels can be then translated into impact-based forecasts.
* * *
2018-26, 2018.